# Privacy Amplification by Subsampling: Tight Analyses via Couplings and Divergences

**Borja Balle**
Amazon Research
pigem@amazon.co.uk

**Gilles Barthe**
IMDEA Software Institute
gilles.barthe@imdea.org

**Marco Gaboardi**
University at Buffalo, SUNY
gaboardi@buffalo.edu

## Abstract

Differential privacy comes equipped with multiple analytical tools for the design of private data analyses. One important tool is the so-called "privacy amplification by subsampling" principle, which ensures that a differentially private mechanism run on a random subsample of a population provides higher privacy guarantees than when run on the entire population. Several instances of this principle have been studied for different random subsampling methods, each with an ad-hoc analysis. In this paper we present a general method that recovers and improves prior analyses, yields lower bounds and derives new instances of privacy amplification by subsampling. Our method leverages a characterization of differential privacy as a divergence which emerged in the program verification community. Furthermore, it introduces new tools, including advanced joint convexity and privacy profiles, which might be of independent interest.

## 1 Introduction

Subsampling is a fundamental tool in the design and analysis of differentially private mechanisms. Broadly speaking, the intuition behind the "privacy amplification by subsampling" principle is that the privacy guarantees of a differentially private mechanism can be amplified by applying it to a small random subsample of records from a given dataset. In machine learning, many classes of algorithms involve sampling operations, e.g. stochastic optimization methods and Bayesian inference algorithms, and it is not surprising that results quantifying the privacy amplification obtained via subsampling play a key role in designing differentially private versions of these learning algorithms [Bassily et al., 2014, Wang et al., 2015, Abadi et al., 2016, Jälkö et al., 2017, Park et al., 2016b,a]. Additionally, from a practical standpoint subsampling provides a straightforward method to obtain privacy amplification when the final mechanism is only available as a black-box. For example, in Apple's iOS and Google's Chrome deployments of differential privacy for data collection the privacy parameters are hard-coded into the implementation and cannot be modified by the user. In this type of settings, if the default privacy parameters are not satisfactory one could achieve a stronger privacy guarantee by devising a strategy that only submits to the mechanism a random sample of the data.

Despite the practical importance of subsampling, existing tools to bound privacy amplification only work for specific forms of subsampling and typically come with cumbersome proofs providing no information about the tightness of the resulting bounds. In this paper we remedy this situation by providing a general framework for deriving tight privacy amplification results that can be applied to any of the subsampling strategies considered in the literature. Our framework builds on a characterization of differential privacy in terms of $\alpha$-divergences [Barthe and Olmedo, 2013]. This characterization has been used before for program verification [Barthe et al., 2012, 2016], while we use it here for the first time in the context of algorithm analysis. In order to do this, we develop several novel analytical tools, including *advanced joint convexity* – a property of $\alpha$-divergence with respect to

mixture distributions – and *privacy profiles* – a general tool describing the privacy guarantees that private algorithms provide.

One of our motivations to initiate a systematic study of privacy amplification by subsampling is that this is an important primitive for the design of differentially private algorithms which has received less attention than other building blocks like composition theorems [Dwork et al., 2010, Kairouz et al., 2017, Murtagh and Vadhan, 2016]. Given the relevance of sampling operations in machine learning, it is important to understand what are the limitations of privacy amplification and develop a fine-grained understanding of its theoretical properties. Our results provide a first step in this direction by showing how privacy amplification resulting from different sampling techniques can be analyzed by means of single set of tools, and by showing how these tools can be used for proving lower bounds. Our analyses also highlight the importance of choosing a sampling technique that is well-adapted to the notion of neighbouring datasets under consideration. A second motivation is that subsampling provides a natural example of mechanisms where the output distribution is a mixture. Because mixtures have an additive structure and differential privacy is defined in terms of a multiplicative guarantee, analyzing the privacy guarantees of mechanisms whose output distribution is a mixture is in general a challenging task. Although our analyses are specialized to mixtures arising from subsampling, we believe the tools we develop in terms of couplings and divergences will also be useful to analyze other types of mechanisms involving mixture distributions. Finally, we want to remark that privacy amplification results also play a role in analyzing the generalization and sample complexity properties of private learning algorithms [Kasiviswanathan et al., 2011, Beimel et al., 2013, Bun et al., 2015, Wang et al., 2016]; an in-depth understanding of the interplay between sampling and differential privacy might also have applications in this direction.

## 2  Problem Statement and Methodology Overview

A *mechanism* $\mathcal{M} : X \to \mathbb{P}(Z)$ with input space $X$ and output space $Z$ is a randomized algorithm that on input $x$ outputs a sample from the distribution $\mathcal{M}(x)$ over $Z$. Here $\mathbb{P}(Z)$ denotes the set of probability measures on the output space $Z$. We implicitly assume $Z$ is equipped with a sigma-algebra of measurable subsets and a base measure, in which case $\mathbb{P}(Z)$ is restricted to probability measures that are absolutely continuous with respect to the base measure. In most cases of interest $Z$ is either a discrete space equipped with the counting measure or an Euclidean space equipped with the Lebesgue measure. We also assume $X$ is equipped with a binary symmetric relation $\simeq_X$ defining the notion of neighbouring inputs.

Let $\varepsilon \geq 0$ and $\delta \in [0, 1]$. A mechanism $\mathcal{M}$ is said to be $(\varepsilon, \delta)$-*differentially private* w.r.t. $\simeq_X$ if for every pair of inputs $x \simeq_X x'$ and every measurable subset $E \subseteq Z$ we have

$$\Pr[\mathcal{M}(x) \in E] \leq e^\varepsilon \Pr[\mathcal{M}(x') \in E] + \delta \ . \tag{1}$$

For our purposes, it will be more convenient to express differential privacy in terms of $\alpha$-divergences[1]. Concretely, the $\alpha$-divergence ($\alpha \geq 1$) between two probability measures $\mu, \mu' \in \mathbb{P}(Z)$ is defined as[2]

$$D_\alpha(\mu \| \mu') = \sup_E \left( \mu(E) - \alpha \mu'(E) \right) = \int_Z \left[ \frac{d\mu}{d\mu'}(z) - \alpha \right]_+ d\mu'(z) = \sum_{z \in Z} [\mu(z) - \alpha \mu'(z)]_+ \ , \tag{2}$$

where $E$ ranges over all measurable subsets of $Z$, $[\bullet]_+ = \max\{\bullet, 0\}$, and the last equality is a specialization for discrete $Z$. It is easy to see [Barthe and Olmedo, 2013] that $\mathcal{M}$ is $(\varepsilon, \delta)$-differentially private if and only if $D_{e^\varepsilon}(\mathcal{M}(x) \| \mathcal{M}(x')) \leq \delta$ for every $x$ and $x'$ such that $x \simeq_X x'$.

In order to emphasize the relevant properties of $\mathcal{M}$ from a privacy amplification point of view, we introduce the concepts of *privacy profile* and *group-privacy profiles*. The privacy profile $\delta_\mathcal{M}$ of a mechanism $\mathcal{M}$ is a function associating to each privacy parameter $\alpha = e^\varepsilon$ a bound on the $\alpha$-divergence between the results of running the mechanism on two adjacent datasets, i.e. $\delta_\mathcal{M}(\varepsilon) = \sup_{x \simeq_X x'} D_{e^\varepsilon}(\mathcal{M}(x) \| \mathcal{M}(x'))$ (we will discuss the properties of this tool in more details in the next section). Informally speaking, the privacy profile represents the set of all of privacy parameters under

which a mechanism provides differential privacy. In particular, recall that an $(\varepsilon, \delta)$-DP mechanism $\mathcal{M}$ is also $(\varepsilon', \delta')$-DP for any $\varepsilon' \geq \varepsilon$ and any $\delta' \geq \delta$. The privacy profile $\delta_{\mathcal{M}}$ defines a curve in $[0, \infty) \times [0, 1]$ that separates the space of privacy parameters into two regions: the ones for which $\mathcal{M}$ satisfies differential privacy and the ones for which it does not. This curve exists for every mechanism $\mathcal{M}$, even for mechanisms that satisfy pure DP for some value of $\varepsilon$. When the mechanism is clear from the context we might slightly abuse our notation and write $\delta(\varepsilon)$ or $\delta$ for the corresponding privacy profile. To define group-privacy profiles $\delta_{\mathcal{M},k}$ $(k \geq 1)$ we use the path-distance $d$ induced by $\simeq_X$:

$$d(x, x') = \min\{k : \exists x_1, \ldots, x_{k-1}, x \simeq_X x_1, x_1 \simeq_X x_2, \ldots, x_{k-1} \simeq_X x'\} \ .$$

With this notation, we define $\delta_{\mathcal{M},k}(\varepsilon) = \sup_{d(x,x') \leq k} D_{e^\varepsilon}(\mathcal{M}(x) \| \mathcal{M}(x'))$. Note that $\delta_{\mathcal{M}} = \delta_{\mathcal{M},1}$.

**Problem Statement**  A well-known method for increasing privacy of a mechanism is to apply the mechanism to a random subsample of the input database, rather than on the database itself. Intuitively, the method decreases the chances of leaking information about a particular individual because nothing about that individual can be leaked in the cases where the individual is not included in the subsample. The question addressed in this paper is to devise methods for quantifying amplification and for proving optimality of the bounds. This turns out to be a surprisingly subtle problem.

Formally, let $X$ and $Y$ be two sets equipped with neighbouring relations $\simeq_X$ and $\simeq_Y$ respectively. We assume that both $X$ and $Y$ contain databases (modelled as sets, multisets, or tuples) over a universe $\mathcal{U}$ that represents all possible records contained in a database. A subsampling mechanism is a randomized algorithm $\mathcal{S} : X \to \mathbb{P}(Y)$ that takes as input a database $x$ and outputs a finitely supported distribution over datasets. Note that we find it convenient to distinguish between $X$ and $Y$ because $x$ and $y$ might not always have the same type. For example, sampling with replacement from a set $x$ yields a multiset $y$.

The problem of privacy amplification can now be stated as follows: let $\mathcal{M} : Y \to \mathbb{P}(Z)$ be a mechanism with privacy profile $\delta_{\mathcal{M}}$ with respect to $\simeq_Y$, and let $\mathcal{S}$ be a subsampling mechanism. Consider the subsampled mechanism $\mathcal{M}^{\mathcal{S}} : X \to \mathbb{P}(Z)$ given by $\mathcal{M}^{\mathcal{S}}(x) = \mathcal{M}(\mathcal{S}(x))$, where the composition notation means we feed a sample from $\mathcal{S}(x)$ into $\mathcal{M}$. The goal is to relate the privacy profiles of $\mathcal{M}$ and $\mathcal{M}^{\mathcal{S}}$, via an inequality of the form: for every $\varepsilon \geq 0$, there exists $0 \leq \varepsilon' \leq \varepsilon$ such that $\delta_{\mathcal{M}^{\mathcal{S}}}(\varepsilon') \leq h(\delta_{\mathcal{M}}(\varepsilon))$, where $h$ is some function to be determined. In terms of differential privacy, one can be read as saying that if $\mathcal{M}$ is $(\varepsilon, \delta)$-DP, then the subsampled mechanism $\mathcal{M}^{\mathcal{S}}$ is $(\varepsilon', h(\delta))$-DP for some $\varepsilon' \leq \varepsilon$. This is a privacy amplification statement because the new mechanism has better privacy parameters than the original one.

A full specification of this problem requires formalizing the following three ingredients: (i) *dataset representation* specifying whether the inputs to the mechanism are sets, multisets, or tuples; (ii) *neighbouring relations* in $X$ and $Y$, including the usual remove/add-one $\simeq_r$ and substitute-one $\simeq_s$ relations; (iii) *subsampling method* and its parameters, with the most commonly used being subsample without replacement, subsampling with replacement, and Poisson subsampling.

Regardless of the specific setting being considered, the main challenge in the analysis of privacy amplification by subsampling resides in the fact that the output distribution of the mechanism $\mu = \mathcal{M}^{\mathcal{S}}(x) \in \mathbb{P}(Z)$ is a *mixture distribution*. In particular, writing $\mu_y = \mathcal{M}(y) \in \mathbb{P}(Z)$ for any $y \in Y$ and taking $\omega = \mathcal{S}(x) \in \mathbb{P}(Y)$ to be the (finitely supported) distribution over subsamples from $x$ produced by the subsampling mechanism, we can write $\mu = \sum_y \omega(y)\mu_y = \omega M$, where $M$ denotes the Markov kernel operating on measures defined by $\mathcal{M}$. Consequently, proving privacy amplifications results requires reasoning about the mixtures obtained when sampling from two neighbouring datasets $x \simeq_X x'$, and how the privacy parameters are affected by the mixture.

**Our Contribution**  We provide a unified method for deriving privacy amplification by subsampling bounds (Section 3). Our method recovers all existing results in the literature and allow us to derive novel amplification bounds (Section 4). In most cases our method also provides optimal constants which are shown to be tight by a generic lower bound (Section 5). Our analysis relies on properties of divergences and privacy profiles, together with two additional ingredients.

The first ingredient is a novel *advanced joint convexity* property providing upper bounds on the $\alpha$-divergence between overlapping mixture distributions. In the specific context of differential privacy this result yields for every $x \simeq_X x'$:

$$D_{e^{\varepsilon'}}(\mathcal{M}^{\mathcal{S}}(x) \| \mathcal{M}^{\mathcal{S}}(x')) \leq \eta \cdot ((1 - \beta)D_{e^\varepsilon}(\mu_1 \| \mu_0) + \beta D_{e^\varepsilon}(\mu_1 \| \mu_1')) \ , \tag{3}$$

| Subsampling | $\simeq_Y$ | $\simeq_X$ | $\eta$ | $\delta'$ | Theorem |
|---|---|---|---|---|---|
| Poisson($\gamma$) | R | R | $\gamma$ | $\gamma\delta$ | 8 |
| WOR($n,m$) | S | S | $\frac{m}{n}$ | $\frac{m}{n}\delta$ | 9 |
| WR($n,m$) | S | S | $1-\left(1-\frac{1}{n}\right)^m$ | $\sum_{k=1}^m \binom{m}{k}\left(\frac{1}{n}\right)^k\left(1-\frac{1}{n}\right)^{m-k}\delta_k$ | 10 |
| WR($n,m$) | S | R | $1-\left(1-\frac{1}{n}\right)^m$ | $\sum_{k=1}^m \binom{m}{k}\left(\frac{1}{n}\right)^k\left(1-\frac{1}{n}\right)^{m-k}\delta_k$ | 11 |

Table 1: Summary of privacy amplification bounds. Amplification parameter $\eta$: $e^{\varepsilon'} = 1 + \eta(e^\varepsilon - 1)$. Types of subsampling: without replacement (WOR) and with replacement (WR). Neighbouring relations: remove/add-one (R) and substitute one (S).

for $e^{\varepsilon'} = 1 + \eta(e^\varepsilon - 1)$, some $\beta \in [0,1]$, and $\eta = \mathsf{TV}(\mathcal{S}(x), \mathcal{S}(x'))$ being the total variation distance between the distributions over subsamples. Here $\mu_0, \mu_1, \mu_1' \in \mathbb{P}(Z)$ are suitable measures obtained from $\mathcal{M}^{\mathcal{S}}(x)$ and $\mathcal{M}^{\mathcal{S}}(x')$ through a coupling and projection operation. In particular, the proof of advanced joint convexity uses ideas from probabilistic couplings, and more specifically the maximal coupling construction (see Theorem 2 and its proof for more details). It is also interesting to note that the non-linear relation $\varepsilon' = \log(1 + \eta(e^\varepsilon - 1))$ already appears in some existing privacy amplification results (e.g. Li et al. [2012]). Although for small $\varepsilon$ and $\eta$ this relation yields $\varepsilon' = O(\eta\varepsilon)$, our results show that the more complicated non-linear relation is in fact a fundamental aspect of privacy amplification by subsampling.

The second ingredient in our analysis establishes an upper bound for the divergences occurring in the right hand side of (3) in terms of group-privacy profiles. It states that under suitable conditions, we have $D_{e^\varepsilon}(\nu M \| \nu' M) \leq \sum_{k\geq 1} \lambda_k(\nu)\delta_{\mathcal{M},k}(e^\varepsilon)$ for suitable choices of $\lambda_k$. Again, the proof of the inequality uses tools from probabilistic couplings.

The combination of these results yields a bound of the privacy profile of $\mathcal{M}^{\mathcal{S}}$ as a function of the group-privacy profiles of $\mathcal{M}$. Based on this inequality, we will establish several privacy amplification result and prove tightness results. This methodology can be applied to any of the settings discussed above in terms of dataset representation, neighbouring relation, and type of subsampling. Table 1 summarizes several results that can be obtained with our method (see Section 4 for details). The supplementary material also contains plots illustrating our bounds (Figure 1) and proofs of all the results presented in the paper.

## 3 Tools: Couplings, Divergences and Privacy Profiles

We next introduce several tools that will be used to support our analyses. The first and second tools are known, whereas the remaining tools are new and of independent interest.

**Divergences** The following characterization follows immediately from the definition of $\alpha$-divergence in terms of the supremum over $E$.

**Theorem 1** ([Barthe and Olmedo, 2013]). *A mechanism $\mathcal{M}$ is $(\varepsilon, \delta)$-differentially private with respect to $\simeq_X$ if and only if $\sup_{x \simeq_X x'} D_{e^\varepsilon}(\mathcal{M}(x) \| \mathcal{M}(x')) \leq \delta$.*

Note that in the statement of the theorem we take $\alpha = e^\varepsilon$. Throughout the paper we sometimes use these two notations interchangeably to make expressions more compact.

We now state consequences of the definition of $\alpha$-divergence: (i) $0 \leq D_\alpha(\mu \| \mu') \leq 1$; (ii) the function $\alpha \mapsto D_\alpha(\mu \| \mu')$ is monotonically decreasing; (iii) the function $(\mu, \mu') \mapsto D_\alpha(\mu \| \mu')$ is jointly convex. Furthermore, one can show that $\lim_{\alpha \to \infty} D_\alpha(\mu \| \mu') = 0$ if and only if $\mathsf{supp}(\mu) \subseteq \mathsf{supp}(\mu')$.

**Couplings** Couplings are a standard tool for deriving upper bounds for the statistical distance between distributions. Concretely, it is well-known that the total variation distance between two distributions $\nu, \nu' \in \mathbb{P}(Y)$ satisfies $\mathsf{TV}(\nu, \nu') \leq \mathsf{Pr}_\pi[y \neq y']$ for any coupling $\pi$, where equality is attained by taking the so-called maximal coupling. We recall the definition of coupling and provide a construction of the maximal coupling, which we shall use in later sections.

A coupling between two distributions $\nu, \nu' \in \mathbb{P}(Y)$ is a distribution $\pi \in \mathbb{P}(Y \times Y)$ whose marginals along the projections $(y, y') \mapsto y$ and $(y, y') \mapsto y'$ are $\nu$ and $\nu'$ respectively. Couplings always exist, and furthermore, there exists a maximal coupling, which exactly characterizes the total variation distance between $\nu$ and $\nu'$. Let $\nu_0(y) = \min\{\nu(y), \nu'(y)\}$ and let $\eta = \mathsf{TV}(\nu, \nu') = 1 - \sum_{y \in Y} \nu_0(y)$, where TV denotes the total variation distance. The maximal coupling between $\nu$ and $\nu'$ is defined as the mixture $\pi = (1 - \eta)\pi_0 + \eta\pi_1$, where $\pi_0(y, y') = \nu_0(y)\mathbb{1}[y = y']/(1 - \eta)$, and $\nu_1(y, y') = (\nu(y) - \nu_0(y))(\nu'(y') - \nu_0(y'))/\eta$. Projecting the maximal coupling along the marginals yields the overlapping mixture decompositions $\nu = (1 - \eta)\nu_0 + \eta\nu_1$ and $\nu' = (1 - \eta)\nu_0 + \eta\nu'_1$.

**Advanced Joint Convexity**   The privacy amplification phenomenon is tightly connected to an interesting new form of joint convexity for $\alpha$-divergences, which we call advanced joint convexity.

**Theorem 2** (Advanced Joint Convexity of $D_\alpha{}^3$). *Let $\mu, \mu' \in \mathbb{P}(Z)$ be measures satisfying $\mu = (1 - \eta)\mu_0 + \eta\mu_1$ and $\mu' = (1 - \eta)\mu_0 + \eta\mu'_1$ for some $\eta$, $\mu_0$, $\mu_1$, and $\mu'_1$. Given $\alpha \geq 1$, let $\alpha' = 1 + \eta(\alpha - 1)$ and $\beta = \alpha'/\alpha$. Then the following holds:*

$$D_{\alpha'}(\mu \| \mu') = \eta D_\alpha(\mu_1 \| (1 - \beta)\mu_0 + \beta\mu'_1) \ . \tag{4}$$

Note that writing $\alpha = e^\varepsilon$ and $\alpha' = e^{\varepsilon'}$ in the above lemma we get the relation $\varepsilon' = \log(1 + \eta(e^\varepsilon - 1))$. Applying standard joint convexity to the right hand side above we conclude: $D_{\alpha'}(\mu \| \mu') \leq (1 - \beta)\eta D_\alpha(\mu_1 \| \mu_0) + \beta\eta D_\alpha(\mu_1 \| \mu'_1)$. Note that applying joint convexity directly on $D_{\alpha'}(\mu \| \mu')$ instead of advanced joint complexity yields a weaker bound which implies amplification for the $\delta$ privacy parameter, but not for the $\varepsilon$ privacy parameter.

When using advanced joint convexity to analyze privacy amplification we consider two elements $x$ and $x'$ and fix the following notation. Let $\omega = \mathcal{S}(x)$ and $\omega' = \mathcal{S}(x')$ and $\mu = \omega M$ and $\mu' = \omega' M$, where we use the notation $M$ to denote the Markov kernel associated with mechanism $\mathcal{M}$ operating on measures over $Y$. We then consider the mixture factorization of $\omega$ and $\omega'$ obtained by taking the decompositions induced by projecting the *maximal coupling* $\pi = (1 - \eta)\pi_0 + \eta\pi_1$ on the first and second marginals: $\omega = (1 - \eta)\omega_0 + \eta\omega_1$ and $\omega' = (1 - \eta)\omega_0 + \eta\omega'_1$. It is easy to see from the construction of the maximal coupling that $\omega_1$ and $\omega'_1$ have disjoint supports and $\eta$ is the smallest probability such that this condition holds. In this way we obtain the canonical mixture decompositions $\mu = (1 - \eta)\mu_0 + \eta\mu_1$ and $\mu' = (1 - \eta)\mu_0 + \eta\mu'_1$, where $\mu_0 = \omega_0 M$, $\mu_1 = \omega_1 M$ and $\mu'_1 = \omega'_1 M$.

**Privacy Profiles**   We state some important properties of privacy profiles. Our first result illustrates our claim that the "privacy curve" exists for every mechanism $\mathcal{M}$ in the context of the Laplace output perturbation mechanism.

**Theorem 3.** *Let $f : X \to \mathbb{R}$ be a function with global sensitivity $\Delta = \sup_{x \simeq_X x'} |f(x) - f(x')|$. Suppose $\mathcal{M}(x) = f(x) + \mathsf{Lap}(b)$ is a Laplace output perturbation mechanism with noise parameter $b$. The privacy profile of $\mathcal{M}$ is given by $\delta_\mathcal{M}(\varepsilon) = [1 - \exp(\frac{\varepsilon - \theta}{2})]_+$, where $\theta = \Delta/b$.*

The well-known fact that the Laplace mechanism with $b \geq \Delta/\varepsilon$ is $(\varepsilon, 0)$-DP follows from this result by noting that $\delta_\mathcal{M}(\varepsilon) = 0$ for any $\varepsilon \geq \theta$. However, Theorem 3 also provides more information: it shows that for $\varepsilon < \Delta/b$ the Laplace mechanism with noise parameter $b$ satisfies $(\varepsilon, \delta)$-DP with $\delta = \delta_\mathcal{M}(\varepsilon)$.

For mechanisms that only satisfy approximate DP, the privacy profile provides information about the behaviour of $\delta_\mathcal{M}(\varepsilon)$ as we increase $\varepsilon \to \infty$. The classical analysis for the Gaussian output perturbation mechanism provides some information in this respect. Recall that for a function $f : X \to \mathbb{R}^d$ with $L_2$ global sensitivity $\Delta = \sup_{x \simeq_X x'} \|f(x) - f(x)\|_2$ the mechanism $\mathcal{M}(x) = f(x) + \mathcal{N}(0, \sigma^2 I)$ satisfies $(\varepsilon, \delta)$-DP if $\sigma^2 \geq 2\Delta^2 \log(1.25/\delta)/\varepsilon^2$ and $\varepsilon \in (0, 1)$ (cf. [Dwork and Roth, 2014, Theorem A.1]). This can be rewritten as $\delta_\mathcal{M}(\varepsilon) \leq 1.25 e^{-\varepsilon^2/2\theta^2}$ for $\varepsilon \in (0, 1)$, where $\theta = \Delta/\sigma$. Recently, [Balle and Wang, 2018] gave a new analysis of the Gaussian mechanism that is valid for all values of $\varepsilon$. Their analysis can be interpreted as providing an expression for the privacy profile of the Gaussian mechanism in terms of the CDF of a standard normal distribution $\Phi(t) = (2\pi)^{-1/2} \int_{-\infty}^t e^{-r^2/2} dr$.

**Theorem 4** ([Balle and Wang, 2018]). *Let $f : X \to \mathbb{R}^d$ be a function with $L_2$ global sensitivity $\Delta$. For any $\sigma > 0$ let $\theta = \Delta/\sigma$. The privacy profile of the Gaussian mechanism $\mathcal{M}(x) = f(x) + \mathcal{N}(0, \sigma^2 I)$ is given by $\delta_\mathcal{M}(e^\varepsilon) = \Phi(\theta/2 - \varepsilon/\theta) - e^\varepsilon \Phi(-\theta/2 - \varepsilon/\theta)$.*

Interestingly, the proof of Theorem 4 implicitly provides a characterization of privacy profiles in terms of privacy loss random variables that holds for any mechanism. Recall that the *privacy loss random variable* of a mechanism $\mathcal{M}$ on inputs $x \simeq_X x'$ is defined as $L_{\mathcal{M}}^{x,x'} = \log(d\mu/d\mu')(\mathbf{z})$, where $\mu = \mathcal{M}(x)$, $\mu' = \mathcal{M}(x')$, and $\mathbf{z} \sim \mu$.

**Theorem 5** ([Balle and Wang, 2018]). *The privacy profile of any mechanism $\mathcal{M}$ satisfies*

$$\delta_{\mathcal{M}}(\varepsilon) = \sup_{x \simeq_X x'} \left( \Pr[L_{\mathcal{M}}^{x,x'} > \varepsilon] - e^{\varepsilon} \Pr[L_{\mathcal{M}}^{x',x} < -\varepsilon] \right) \ .$$

The characterization above generalizes the well-known inequality $\delta_{\mathcal{M}}(\varepsilon) \leq \sup_{x \simeq_X x'} \Pr[L_{\mathcal{M}}^{x,x'} > \varepsilon]$ (eg. see [Dwork and Roth, 2014]). This bound is often used to derive $(\varepsilon, \delta)$-DP guarantees from other notions of privacy defined in terms of the moment generating function of the privacy loss random variable, including concentrated DP [Dwork and Rothblum, 2016], zero-concentrated DP [Bun and Steinke, 2016], Rényi DP [Mironov, 2017], and truncated concentrated DP [Bun et al., 2018]. We now show a reverse implication also holds. Namely, that privacy profiles can be used to recover all the information provided by the moment generating function of the privacy loss random variable.

**Theorem 6.** *Given a mechanism $\mathcal{M}$ and inputs $x \simeq_X x'$ let $\mu = \mathcal{M}(x)$ and $\mu' = \mathcal{M}(x')$. For $s \geq 0$, define the moment generating function $\varphi_{\mathcal{M}}^{x,x'}(s) = \mathsf{E}[\exp(s L_{\mathcal{M}}^{x,x'})]$. Then we have*

$$\varphi_{\mathcal{M}}^{x,x'}(s) = 1 + s(s+1) \int_0^{\infty} \left( e^{s\varepsilon} D_{e^\varepsilon}(\mu\|\mu') + e^{-(s+1)\varepsilon} D_{e^\varepsilon}(\mu'\|\mu) \right) d\varepsilon \ .$$

*In particular, if $D_{e^\varepsilon}(\mu\|\mu') = D_{e^\varepsilon}(\mu'\|\mu)$ holds[4] for every $x \simeq_X x'$, then $\sup_{x \simeq_X x'} \varphi_{\mathcal{M}}^{x,x'}(s) = 1 + s(s+1) \int_0^{\infty} (e^{s\varepsilon} + e^{-(s+1)\varepsilon}) \delta_{\mathcal{M}}(\varepsilon) d\varepsilon$.*

**Group-privacy Profiles** Recall the $k$th group privacy profile of a mechanism $\mathcal{M}$ is defined as $\delta_{\mathcal{M},k}(\varepsilon) = \sup_{d(x,x') \leq k} D_{e^\varepsilon}(\mathcal{M}(x)\|\mathcal{M}(x'))$. A standard group privacy analysis[5] immediately yields $\delta_{\mathcal{M},k}(\varepsilon) \leq (e^\varepsilon - 1)\delta_{\mathcal{M}}(\varepsilon/k)/(e^{\varepsilon/k} - 1)$. However, "white-box" approaches based on full knowledge of the privacy profile of $\mathcal{M}$ can be used to improve this result for specific mechanisms. For example, it is not hard to see that, combining the expressions from Theorems 3 and 4 with the triangle inequality on the global sensitivity of changing $k$ records in a dataset, one obtains bounds that improve on the "black-box" approach for all ranges of parameters for the Laplace and Gaussian mechanisms. This is one of the reasons why we state our bounds directly in terms of (group-)privacy profiles (a numerical comparison can be found in the supplementary material).

**Distance-compatible Coupling** The last tool we need to prove general privacy amplification bounds based on $\alpha$-divergences is the existence of a certain type of couplings between two distributions like the ones occurring in the right hand side of (4). Recall that any coupling $\pi$ between two distributions $\nu, \nu' \in \mathbb{P}(Y)$ can be used to rewrite the mixture distributions $\tilde{\mu} = \nu M$ and $\tilde{\mu}' = \nu' M$ as $\tilde{\mu} = \sum_{y,y'} \pi_{y,y'} \mathcal{M}(y)$ and $\tilde{\mu}' = \sum_{y,y'} \pi_{y,y'} \mathcal{M}(y')$. Using the joint convexity of $D_\alpha$ and the definition of group-privacy profiles to get the bound

$$D_{e^\varepsilon}(\tilde{\mu}\|\tilde{\mu}') \leq \sum_{y,y'} \pi_{y,y'} D_{e^\varepsilon}(\mathcal{M}(y)\|\mathcal{M}(y')) \leq \sum_{y,y'} \pi_{y,y'} \delta_{\mathcal{M},d_Y(y,y')}(\varepsilon) \ . \tag{5}$$

Since this bound holds for any coupling $\pi$, one can set out to optimize it by finding a coupling the minimizes the right hand side of (5). We show that the existence of couplings whose support is contained inside a certain subset of $Y \times Y$ is enough to obtain an optimal bound. Furthermore, we show that when this condition is satisfied the resulting bound depends only on $\nu$ and the group-privacy profiles of $\mathcal{M}$.

We say that two distributions $\nu, \nu' \in \mathbb{P}(Y)$ are $d_Y$-*compatible* if there exists a coupling $\pi$ between $\nu$ and $\nu'$ such for any $(y, y') \in \mathsf{supp}(\pi)$ we have $d_Y(y,y') = d_Y(y, \mathsf{supp}(\nu'))$, where the distance between a point $y$ and the set $\mathsf{supp}(\nu')$ is defined as the distance between $y$ and the closest point in $\mathsf{supp}(\nu')$.

**Theorem 7.** *Let $C(\nu, \nu')$ be the set of all couplings between $\nu$ and $\nu'$ and for $k \geq 1$ let $Y_k = \{y \in \text{supp}(\nu) : d_Y(y, \text{supp}(\nu')) = k\}$. If $\nu$ and $\nu'$ are $d_Y$-compatible, then the following holds:*

$$\min_{\pi \in C(\nu, \nu')} \sum_{y, y'} \pi_{y, y'} \delta_{\mathcal{M}, d_Y(y, y')}(\varepsilon) = \sum_{k \geq 1} \nu(Y_k) \delta_{\mathcal{M}, k}(\varepsilon) \ . \tag{6}$$

Applying this result to the bound resulting from the right hand side of (4) yields most of the concrete privacy amplification results presented in the next section.

## 4 Privacy Amplification Bounds

In this section we provide explicit privacy amplification bounds for the most common subsampling methods and neighbouring relations found in the literature on differential privacy, and provide pointers to existing bounds and other related work. For our analysis we work with order-independent representations of datasets without repetitions, i.e. sets. This is mostly for technical convenience, since all our results also hold if one considers datasets represented as tuples or multisets. Note however that subsampling with replacement for a set can yield a multiset; hence we introduce suitable notations for sets and multisets.

Fix a universe of records $\mathcal{U}$ and let $2 = \{0, 1\}$. We write $2^{\mathcal{U}}$ and $\mathbb{N}^{\mathcal{U}}$ for the spaces of all sets and multisets with records from $\mathcal{U}$. Note every set is also a multiset. For $n \geq 0$ we also write $2_n^{\mathcal{U}}$ and $\mathbb{N}_n^{\mathcal{U}}$ for the spaces of all sets and multisets containing exactly $n$ records[6] from $\mathcal{U}$. Given $x \in \mathbb{N}^{\mathcal{U}}$ we write $x_u$ for the number of occurrences of $u \in \mathcal{U}$ in $x$. The support of a multiset $x$ is the defined as the set $\text{supp}(x) = \{u \in \mathcal{U} : x_u > 0\}$ of elements that occur at least once in $x$. Given multisets $x, x' \in \mathbb{N}^{\mathcal{U}}$ we write $x' \subseteq x$ to denote that $x'_u \leq x_u$ for all $u \in \mathcal{U}$.

For order-independent datasets represented as multisets it is natural to consider the two following neighbouring relations. The *remove/add-one* relation is obtained by letting $x \simeq_r x'$ hold whenever $x \subseteq x'$ with $|x| = |x'| - 1$ or $x' \subseteq x$ with $|x| = |x'| + 1$; i.e. $x'$ is obtained by removing or adding a single element to $x$. The *substitute-one* relation is obtained by letting $x \simeq_s x'$ hold whenever $\|x - x'\|_1 = 2$ and $|x| = |x'|$; i.e. $x'$ is obtained by replacing an element in $x$ with a different element from $\mathcal{U}$. Note how $\simeq_r$ relates pairs of datasets with different sizes, while $\simeq_s$ only relates pairs of datasets with the same size.

**Poisson Subsampling** Perhaps the most well-known privacy amplification result refers to the analysis of Poisson subsampling with respect to the remove/add-one relation. In this case the subsampling mechanism $\mathcal{S}_\gamma^{\text{po}} : 2^{\mathcal{U}} \to \mathbb{P}(2^{\mathcal{U}})$ takes a set $x$ and outputs a sample $y$ from the distribution $\omega = \mathcal{S}_\gamma^{\text{po}}(x)$ supported on all set $y \subseteq x$ given by $\omega(y) = \gamma^{|y|}(1 - \gamma)^{|x| - |y|}$. This corresponds to independently adding to $y$ with probability $\gamma$ each element from $x$. Now, given a mechanism $\mathcal{M} : 2^{\mathcal{U}} \to \mathbb{P}(Z)$ with privacy profile $\delta_{\mathcal{M}}$ with respect to $\simeq_r$, we are interested in bounding the privacy profile of the subsampled mechanism $\mathcal{M}^{\mathcal{S}_\gamma^{\text{wo}}}$ with respect to $\simeq_r$.

**Theorem 8.** *Let $\mathcal{M}' = \mathcal{M}^{\mathcal{S}_\gamma^{\text{po}}}$. For any $\varepsilon \geq 0$ we have $\delta_{\mathcal{M}'}(\varepsilon') \leq \gamma \delta_{\mathcal{M}}(\varepsilon)$, where $\varepsilon' = \log(1 + \gamma(e^\varepsilon - 1))$.*

Privacy amplification with Poisson sampling was used in [Chaudhuri and Mishra, 2006, Beimel et al., 2010, Kasiviswanathan et al., 2011, Beimel et al., 2014], which considered loose bounds. A proof of this tight result in terms of $(\varepsilon, \delta)$-DP was first given in [Li et al., 2012]. In the context of the moments accountant technique based on the moment generating function of the privacy loss random variable, [Abadi et al., 2016] provide an amplification result for Gaussian output perturbation mechanisms under Poisson subsampling.

**Sampling Without Replacement** Another known results on privacy amplification corresponds to the analysis of sampling without replacement with respect to the substitution relation. In this case one considers the subsampling mechanism $\mathcal{S}_m^{\text{wo}} : 2_n^{\mathcal{U}} \to \mathbb{P}(2_m^{\mathcal{U}})$ that given a set $x \in 2_n^{\mathcal{U}}$ of size $n$ outputs a sample from the uniform distribution $\omega = \mathcal{S}_m^{\text{wo}}(x)$ over all subsets $y \subseteq x$ of size $m \leq n$. Then, for a given a mechanism $\mathcal{M} : 2_m^{\mathcal{U}} \to \mathbb{P}(Z)$ with privacy profile $\delta_{\mathcal{M}}$ with respect to the substitution relation $\simeq_s$ on sets of size $m$, we are interested in bounding the privacy profile of the mechanism $\mathcal{M}^{\mathcal{S}_m^{\text{wo}}}$ with respect to the substitution relation on sets of size $n$.

**Theorem 9.** *Let $\mathcal{M}' = \mathcal{M}^{\mathcal{S}_m^{\mathrm{wo}}}$. For any $\varepsilon \geq 0$ we have $\delta_{\mathcal{M}'}(\varepsilon') \leq (m/n)\delta_{\mathcal{M}}(\varepsilon)$, where $\varepsilon' = \log(1 + (m/n)(e^\varepsilon - 1))$.*

This setting has been used in [Beimel et al., 2013, Bassily et al., 2014, Wang et al., 2016] with non-tight bounds. A proof of this tight bound formulated in terms of $(\varepsilon, \delta)$-DP can be directly recovered from Ullman's class notes [Ullman, 2017], although the stated bound is weaker. Rényi DP amplification bounds for subsampling without replacement were developed in [Wang et al., 2019].

**Sampling With Replacement**    Next we consider the case of sampling with replacement with respect to the substitution relation $\simeq_s$. The subsampling with replacement mechanism $\mathcal{S}_m^{\mathrm{wr}} : 2^{\mathcal{U}} \to \mathbb{P}(\mathbb{N}_m^{\mathcal{U}})$ takes a set $x$ of size $n$ and outputs a sample from the multinomial distribution $\omega = \mathcal{S}_m^{\mathrm{wr}}(x)$ over all multisets $y$ of size $m \leq n$ with $\mathrm{supp}(y) \subseteq x$, given by $\omega(y) = (m!/n^m) \prod_{u \in \mathcal{U}} x_u/(y_u!)$. In this case we suppose the base mechanism $\mathcal{M} : \mathbb{N}_m^{\mathcal{U}} \to \mathbb{P}(Z)$ is defined on multisets and has privacy profile $\delta_{\mathcal{M}}$ with respect to $\simeq_s$. We are interested in bounding the privacy profile of the subsampled mechanism $\mathcal{M}^{\mathcal{S}_m^{\mathrm{wr}}} : 2_n^{\mathcal{U}} \to \mathbb{P}(Z)$ with respect to $\simeq_s$.

**Theorem 10.** *Let $\mathcal{M}' = \mathcal{M}^{\mathcal{S}_m^{\mathrm{wr}}}$. Given $\varepsilon \geq 0$ and $\varepsilon' = \log(1 + (1 - (1 - 1/n)^m)(e^\varepsilon - 1))$ we have*

$$\delta_{\mathcal{M}'}(\varepsilon') \leq \sum_{k=1}^{m} \binom{m}{k} \left(\frac{1}{n}\right)^k \left(1 - \frac{1}{n}\right)^{m-k} \delta_{\mathcal{M},k}(\varepsilon) \ .$$

Note that if $m = \gamma n$, then $1 - (1 - 1/n)^m \approx \gamma$. A version of this bound in terms of $(\varepsilon, \delta)$-DP that implicitly uses the group privacy property can be found in [Bun et al., 2015]. Our bound matches the asymptotics of [Bun et al., 2015] while providing optimal constants and allowing for white-box group privacy bounds.

**Hybrid Neighbouring Relations**    Using our method it is also possible to analyze new settings which have not been considered before. One interesting example occurs when there is a mismatch between the two neighbouring relations arising in the analysis. For example, suppose one knows the group-privacy profiles $\delta_{\mathcal{M},k}$ of a base mechanism $\mathcal{M} : \mathbb{N}_m^{\mathcal{U}} \to \mathbb{P}(Z)$ with respect to the substitution relation $\simeq_s$. In this case one could ask whether it makes sense to study the privacy profile of the subsampled mechanism $\mathcal{M}^{\mathcal{S}_m^{\mathrm{wr}}} : 2^{\mathcal{U}} \to \mathbb{P}(Z)$ with respect to the remove/add relation $\simeq_r$. In principle, this makes sense in settings where the size of the inputs to $\mathcal{M}$ is restricted due to implementation constraints (eg. limited by the memory available in a GPU used to run a private mechanism that computes a gradient on a mini-batch of size $m$). In this case one might still be interested in analyzing the privacy loss incurred from releasing such stochastic gradients under the remove/add relation. Note that this setting cannot be implemented using sampling without replacement since under the remove/add relation we cannot a priori guarantee that the input dataset will have at least size $m$ because the size of the dataset must be kept private [Vadhan, 2017]. Furthermore, one cannot hope to get a meaningful result about the privacy profile of the subsampled mechanism across all inputs sets in $2^{\mathcal{U}}$; instead the privacy guarantee will depend on the size of the input dataset as shown in the following result.

**Theorem 11.** *Let $\mathcal{M}' = \mathcal{M}^{\mathcal{S}_m^{\mathrm{wr}}}$. For any $\varepsilon \geq 0$ and $n \geq 0$ we have*

$$\sup_{x \in 2_n^{\mathcal{U}}, x \simeq_r x'} D_{e^{\varepsilon'}}(\mathcal{M}'(x) \| \mathcal{M}'(x')) \leq \sum_{k=1}^{m} \binom{m}{k} \left(\frac{1}{n}\right)^k \left(1 - \frac{1}{n}\right)^{m-k} \delta_{\mathcal{M},k}(\varepsilon) \ ,$$

*where $\varepsilon' = \log(1 + (1 - (1 - 1/n)^m)(e^\varepsilon - 1))$.*

**When the Neighbouring Relation is "Incompatible"**    Now we consider a simple example where distance-compatible couplings are not available: Poisson subsampling with respect to the substitution relation. Suppose $x, x' \in 2_n^{\mathcal{U}}$ are sets of size $n$ related by the substitution relation $\simeq_s$. Let $\omega = \mathcal{S}_\eta^{\mathrm{po}}(x)$ and $\omega' = \mathcal{S}_\eta^{\mathrm{po}}(x')$ and note that $\mathsf{TV}(\omega, \omega') = \eta$. Let $x_0 = x \cap x'$ and $v = x \setminus x_0$, $v' = x' \setminus x_0$. In this case the factorization induced by the maximal coupling is obtained by taking $\omega_0 = \mathcal{S}_\eta^{\mathrm{po}}(x_0)$, $\omega_1(y \cup \{v\}) = \omega_0(y)$, and $\omega_1'(y \cup \{v'\}) = \omega_0(y)$. Now the support of $\omega_0$ contains sets of sizes between 0 and $n - 1$, while the supports of $\omega_1$ and $\omega_1$ contain sets of sizes between 1 and $n$. From this observation one can deduce that $\omega_1$ and $\omega_0$ are not $d_{\simeq_s}$-compatible, and $\omega_1$ and $\omega_1'$ are not $d_{\simeq_r}$-compatible.

This argument shows that the method we used to analyze the previous settings cannot be extended to analyze Poisson subsampling under the substitution relation, regardless of whether the privacy profile of the base mechanism is given in terms of the replacement/addition or the substitution relation. This observation is saying that some pairings between subsampling method and neighbouring relation are more natural than others. Nonetheless, even without distance-compatible couplings it is possible to provide privacy amplification bounds for Poisson subsampling with respect to the substitution relation, although the resulting bound is quite cumbersome. The corresponding statement and analysis can be found in the supplementary material.

## 5   Lower Bounds

In this section we show that many of the results given in the previous section are tight by constructing a randomized membership mechanism that attains these upper bounds. For the sake of generality, we state the main construction in terms of tuples instead of multisets. In fact, we prove a general lemma that can be used to obtain tightness results for any subsampling mechanism and any neighbouring relation satisfying two natural assumptions.

For $p \in [0, 1]$ let $\mathcal{R}_p : \{0, 1\} \to \mathbb{P}(\{0, 1\})$ be the randomized response mechanism that given $b \in \{0, 1\}$ returns $b$ with probability $p$ and $1 - b$ with probability $1 - p$. Note that for $p = (e^\varepsilon + \delta)/(e^\varepsilon + 1)$ this mechanism is $(\varepsilon, \delta)$-DP. Let $\nu_0 = \mathcal{R}_p(0)$ and $\nu_1 = \mathcal{R}_p(1)$. For any $\varepsilon \geq 0$ and $p \in [0, 1]$ define $\psi_p(\varepsilon) = [p - e^\varepsilon(1 - p)]_+$. It is easy to verify that $D_{e^\varepsilon}(\nu_0 \| \nu_1) = D_{e^\varepsilon}(\nu_1 \| \nu_0) = \psi_p(\varepsilon)$. Now let $\mathcal{U}$ be a universe containing at least two elements. For $v \in \mathcal{U}$ and $p \in [0, 1]$ we define the *randomized membership* mechanism $\mathcal{M}_{v,p}$ that given a tuple $x = (u_1, \ldots, u_n) \in \mathcal{U}^\star$ returns $\mathcal{M}_{v,p}(x) = \mathcal{R}_p(\mathbb{I}[v \in x])$. We say that a subsampling mechanism $\mathcal{S} : X \to \mathbb{P}(\mathcal{U}^\star)$ defined on some set $X \subseteq \mathcal{U}^\star$ is *natural* if the following two conditions are satisfied: (1) for any $x \in X$ and $u \in \mathcal{U}$, if $u \in x$ then there exists $y \in \mathsf{supp}(\mathcal{S}(x))$ such that $u \in y$; (2) for any $x \in X$ and $u \in \mathcal{U}$, if $u \notin x$ then we have $u \notin y$ for every $y \in \mathsf{supp}(\mathcal{S}(x))$.

**Lemma 12.** *Let $X \subseteq \mathcal{U}^\star$ be equipped with a neighbouring relation $\simeq_X$ such that there exist $x \simeq_X x'$ with $v \in x$ and $v \notin x'$. Suppose $\mathcal{S} : X \to \mathbb{P}(\mathcal{U}^\star)$ is a natural subsampling mechanism and let $\eta = \sup_{x \simeq_X x'} \mathsf{TV}(\mathcal{S}(x), \mathcal{S}(x'))$. For any $\varepsilon \geq 0$ and $\varepsilon' = \log(1 + \eta(e^\varepsilon - 1))$ we have*

$$\delta_{\mathcal{M}_{v,p}^{\mathcal{S}}}(\varepsilon') = \sup_{x \simeq_X x'} D_{e^{\varepsilon'}}(\mathcal{M}_{v,p}^{\mathcal{S}}(x) \| \mathcal{M}_{v,p}^{\mathcal{S}}(x')) = \eta \psi_p(\varepsilon) \ .$$

We can now apply this lemma to show that the first three results from previous section are tight. This requires specializing from tuples to (multi)sets, and plugging in the definitions of neighbouring relation, subsampling mechanism, and $\eta$ used in each of these theorems.

**Theorem 13.** *The mechanism $\mathcal{M}_{v,p}$ attains the bounds in Theorems 8, 9, 10 for any $p$ and $\eta$.*

## 6   Conclusions

We have developed a general method for reasoning about privacy amplification by subsampling. Our method is applicable to many different settings, some which have already been studied in the literature, and others which are new. Technically, our method leverages two new tools of independent interest: advanced joint convexity and privacy profiles. In the future, it would be interesting to study whether our tools can be extended to give concrete bounds on privacy amplification for other privacy notions such as concentrated DP [Dwork and Rothblum, 2016], zero-concentrated DP [Bun and Steinke, 2016], Rényi DP [Mironov, 2017], and truncated concentrated DP [Bun et al., 2018]. A good starting point is Theorem 6 establishing relations between privacy profiles and moment generating functions of the privacy loss random variable. An alternative approach is to extend the recent results for Rényi DP amplification by subsampling without replacement given in [Wang et al., 2019] to more general notions of subsampling and neighbouring relations.

**Acknowledgments**

This research was initiated during the 2017 Probabilistic Programming Languages workshop hosted by McGill University's Bellairs Research Institute.

## Footnotes

[1]Also known in the literature as *elementary divergences* [Österreicher, 2002] and *hockey-stick divergences* [Sason and Verdú, 2016].

[2]Here $d\mu/d\mu'$ denotes the Radon-Nikodym derivative between $\mu$ and $\mu'$. In particular, if $\mu$ and $\mu'$ have densities $p = d\mu/d\nu$ and $p' = d\mu'/d\nu$ with respect to some base measure $\nu$, then $d\mu/d\mu' = p/p'$.

[3]Proofs of all our results are presented in the appendix.

[4]For example, this is satisfied by all output perturbation mechanisms with symmetric noise distributions.

[5]If $\mathcal{M}$ is $(\varepsilon, \delta)$-DP with respect to $\simeq_Y$, then it is $(k\varepsilon, ((e^{k\varepsilon} - 1)/(e^\varepsilon - 1))\delta)$-DP with respect to $\simeq_Y^k$, cf. [Vadhan, 2017, Lemma 2.2]

[6]In the case of multisets records are counted with multiplicity.

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
