[Supplementary Material]

# A  Plots of Privacy Profiles

(a) Privacy profiles with mechanisms calibrated to provide the same $\delta$ at $\varepsilon = 0$. Profile expressions are given in Section 5 (RR), Theorem 3 (Laplace), and Theorem 4 (Gauss).

(b) Subsampled Gaussian mechanism. Comparison between sampling without replacement (Theorem 9) and with replacement (Theorem 10, with white-box group privacy), both with the same subsampled dataset sizes.

(c) Subsampled Laplace mechanism. Comparison between sampling without replacement (Theorem 9) and with replacement (Theorem 10, with white-box group privacy), both with the same subsampled dataset sizes.

(d) Subsampled Laplace mechanism. Impact of group-privacy effect in sampling with replacement (white-box group privacy).

(e) Subsampled Laplace mechanism. Impact of white-box vs. black-box group-privacy in sampling with replacement.

Figure 1: Plots of privacy profiles. Results illustrate the notion of privacy profile and the different subsampling bounds derived in the paper.

# B Proofs from Section 3

*Proof of Theorem 2.* It suffices to check that for any $z \in Z$,

$$[\mu(z) - \alpha'\mu'(z)]_+ = \eta\,[\mu_1(z) - \alpha\,((1-\beta)\mu_0(z) + \beta\mu_1'(z))]_+ \quad .$$

Plugging this identity in the definition of $D_{\alpha'}$ we get the desired equality

$$D_{\alpha'}(\mu\|\mu') = \eta D_{\alpha}(\mu_1\|(1-\beta)\mu_0 + \beta\mu_1') \quad .$$

$\square$

*Proof of Theorem 3.* Suppose $x \simeq_X x'$ and assume without loss of generality that $y = f(x) = 0$ and $y' = f(x) = \Delta > 0$. Plugging the density of the Laplace distribution in the definition of $\alpha$-divergence we get

$$D_{e^\varepsilon}(\mathsf{Lap}(b)\|\Delta + \mathsf{Lap}(b)) = \frac{1}{2b}\int_{\mathbb{R}}\left[e^{-\frac{|z|}{b}} - e^\varepsilon e^{-\frac{|z-\Delta|}{b}}\right]_+ dz \quad .$$

Now we observe that the quantity inside the integral above is positive if and only if $|z - \Delta| - |z| \geq \varepsilon b$. Since $||z+\Delta| - |z|| \leq \Delta$, we see that the divergence is zero for $\varepsilon > \Delta/b$. On the other hand, for $\varepsilon \in [0, \Delta/b]$ we have $\{z : |z - \Delta| - |z| \geq \varepsilon b\} = (-\infty, (\Delta - \varepsilon b)/2]$. Thus, we have

$$\frac{1}{2b}\int_{\mathbb{R}}\left[e^{-\frac{|z|}{b}} - e^\varepsilon e^{-\frac{|z-\Delta|}{b}}\right]_+ dz = \frac{1}{2b}\int_{-\infty}^{(\Delta-\varepsilon b)/2} e^{-\frac{|z|}{b}} dz - \frac{e^\varepsilon}{2b}\int_{-\infty}^{(\Delta-\varepsilon b)/2} e^{-\frac{|z-\Delta|}{b}} dz \quad .$$

Now we can compute both integrals as probabilities under the Laplace distribution:

$$\frac{1}{2b}\int_{-\infty}^{(\Delta-\varepsilon b)/2} e^{-\frac{|z|}{b}} dz = \Pr\left[\mathsf{Lap}(b) \leq \frac{\Delta - \varepsilon b}{2}\right]$$
$$= 1 - \frac{1}{2}\exp\left(\frac{\varepsilon b - \Delta}{2b}\right) \quad ,$$
$$\frac{e^\varepsilon}{2b}\int_{-\infty}^{(\Delta-\varepsilon b)/2} e^{-\frac{|z-\Delta|}{b}} dz = e^\varepsilon\Pr\left[\mathsf{Lap}(b) \leq \frac{-\Delta - \varepsilon b}{2}\right]$$
$$= \frac{e^\varepsilon}{2}\exp\left(\frac{-\varepsilon b - \Delta}{2b}\right) \quad .$$

Putting these two quantities together we finally get, for $\varepsilon \leq \Delta/b$:

$$D_{e^\varepsilon}(\mathsf{Lap}(b)\|\Delta + \mathsf{Lap}(b)) = 1 - \exp\left(\frac{\varepsilon}{2} - \frac{\Delta}{2b}\right) \quad .$$

$\square$

*Proof of Theorem 6.* Let $\varphi = \varphi_{\mathcal{M}}^{x,x'}$, $L = L_{\mathcal{M}}^{x,x'}$, $\tilde{\varphi} = \varphi_{\mathcal{M}}^{x',x}$, and $\tilde{L} = L_{\mathcal{M}}^{x',x}$. Recall that for any non-negative random variable $\mathbf{z}$ one has $\mathsf{E}[\mathbf{z}] = \int_0^\infty \Pr[\mathbf{z} > t]dt$. We use this to write the moment generating function of the corresponding privacy loss random variable for $s \geq 0$ as follows:

$$\varphi(s) = \int_0^\infty \Pr[e^{sL} > t]dt$$
$$= \int_0^\infty \Pr\left[\frac{p(\mathbf{z})}{q(\mathbf{z})} > t^{1/s}\right] dt \quad ,$$

where $\mathbf{z} \sim \mu$, and $p$ and $q$ represent the densities of $\mu$ and $\nu$ with respect to a fixed base measure. Next we observe the probability inside the integral above can be decomposed in terms of a divergence

and a second integral with respect to $q$:

$$\Pr\left[\frac{p(\mathbf{z})}{q(\mathbf{z})} > t^{1/s}\right] = \Pr[p(\mathbf{z}) > t^{1/s}q(\mathbf{z})]$$

$$= \mathsf{E}_\mu\left[\mathbb{I}[p > t^{1/s}q]\right]$$

$$= \int \mathbb{I}[p(z) > t^{1/s}q(z)]p(z)dz$$

$$= \int \mathbb{I}[p(z) > t^{1/s}q(z)](p(z) - t^{1/s}q(z))dz + t^{1/s}\int \mathbb{I}[p(z) > t^{1/s}q(z)]q(z)dz$$

$$= \int [p(z) - t^{1/s}q(z)]_+ dz + t^{1/s}\int \mathbb{I}[p(z) > t^{1/s}q(z)]q(z)dz$$

$$= D_{t^{1/s}}(\mu\|\mu') + t^{1/s}\int \mathbb{I}[p(z) > t^{1/s}q(z)]q(z)dz \ .$$

Note the term $D_{t^{1/s}}(\mu\|\mu')$ above is not a divergence when $t^{1/s} < 1$. The integral term above can be re-written as a probability in terms of $\tilde{L}$ as follows:

$$\int \mathbb{I}[p(z) > t^{1/s}q(z)]q(z)dz = \Pr[p(\mathbf{z}') > t^{1/s}q(\mathbf{z}')]$$

$$= \Pr\left[\frac{p(\mathbf{z}')}{q(\mathbf{z}')} > t^{1/s}\right]$$

$$= \Pr\left[e^{-\tilde{L}} > t^{1/s}\right] \ ,$$

where $\mathbf{z}' \sim \mu'$. Thus, integrating with respect to $t$ we get an expression for $\varphi(s)$ involving two terms that we will need to massage further:

$$\varphi(s) = \int_0^\infty D_{t^{1/s}}(\mu\|\mu')dt + \int_0^\infty t^{1/s}\Pr\left[e^{-\tilde{L}} > t^{1/s}\right]dt \ .$$

To compute the second integral in the RHS above we perform the change of variables $dt' = t^{1/s}dt$, which comes from taking $t' = t^{1+1/s}/(1+1/s)$, or, equivalently, $t = ((1+1/s)t')^{1/(1+1/s)}$. This allows us to introduce the moment generating function of $\tilde{L}$ as follows:

$$\int_0^\infty t^{1/s}\Pr\left[e^{-\tilde{L}} > t^{1/s}\right]dt = \int_0^\infty \Pr\left[e^{-\tilde{L}} > ((1+1/s)t')^{1/(s+1)}\right]dt'$$

$$= \int_0^\infty \Pr\left[\frac{s}{s+1}e^{-(s+1)\tilde{L}} > t'\right]dt'$$

$$= \frac{s}{s+1}\mathsf{E}\left[e^{-(s+1)\tilde{L}}\right]$$

$$= \frac{s}{s+1}\tilde{\varphi}(-s-1) \ .$$

Putting the derivations above together and substituting $\tilde{\varphi}(-s-1)$ for $\varphi(s)$ we see that

$$\varphi(s) = \frac{s}{s+1}\varphi(s) + \int_0^\infty D_{t^{1/s}}(\mu\|\mu')dt \ ,$$

or equivalently:

$$\varphi(s) = (s+1)\int_0^\infty D_{t^{1/s}}(\mu\|\mu')dt \ .$$

Now we observe that some terms in the integral above cannot be bounded using an $\alpha$-divergence between $\mu$ and $\mu'$, e.g. for $t \in (0,1)$ the term $D_{t^{1/s}}(\mu\|\mu')$ is not a divergence. Instead, using the definition of $D_{t^{1/s}}(\mu\|\mu')$ we can see that these terms are equal to by $1 - t^{1/s} + t^{1/s}D_{t^{-1/s}}(\mu'\|\mu)$,

where the last term is now a divergence. Thus, we split the integral in the expression for $\varphi(s)$ into two parts and obtain

$$\varphi(s) = (s+1)\int_0^1 \left(1 - t'^{1/s} + t'^{1/s}D_{t'^{-1/s}}(\mu'\|\mu)\right)dt' + (s+1)\int_1^\infty D_{t^{1/s}}(\mu\|\mu')dt$$

$$= 1 + (s+1)\int_0^1 t'^{1/s}D_{t'^{-1/s}}(\mu'\|\mu)dt' + (s+1)\int_1^\infty D_{t^{1/s}}(\mu\|\mu')dt \ .$$

Finally, we can obtain the desired equation by performing a series of simple changes of variables $t' = 1/t$, $\alpha = t^{1/s}$, and $\alpha = e^\varepsilon$:

$$\varphi(s) = 1 + (s+1)\int_1^\infty t^{-2-1/s}D_{t^{1/s}}(\mu'\|\mu)dt + (s+1)\int_1^\infty D_{t^{1/s}}(\mu\|\mu')dt$$

$$= 1 + s(s+1)\int_1^\infty \left(\alpha^{s-1}D_\alpha(\mu\|\mu') + \alpha^{-s-2}D_\alpha(\mu'\|\mu)\right)d\alpha$$

$$= 1 + s(s+1)\int_0^\infty \left(e^{s\varepsilon}D_{e^\varepsilon}(\mu\|\mu') + e^{-(s+1)\varepsilon}D_{e^\varepsilon}(\mu'\|\mu)\right)d\varepsilon \ .$$

$\square$

*Proof of Theorem 7.* The result follows from a few simple observations. The first observation is that for any coupling $\pi \in C(\nu,\nu')$ and $y \in \mathsf{supp}(\nu')$ we have

$$\sum_{y'} \pi_{y,y'}\delta_{\mathcal{M},d(y,y')}(\varepsilon) \geq \sum_{y'} \pi_{y,y'}\delta_{\mathcal{M},d(y,\mathsf{supp}(\nu'))}(\varepsilon)$$

$$= \sum_y \nu_y \delta_{\mathcal{M},d(y,\mathsf{supp}(\nu'))}(\varepsilon) \ ,$$

where the first inequality follows from $d(y,y') \geq d(y,\mathsf{supp}(\nu'))$ and the fact that $\delta_{\mathcal{M},k}(\varepsilon)$ is monotonically increasing with $k$. Thus the RHS of (6) is always a lower bound for the LHS. Now let $\pi$ be a $d_Y$-compatible coupling. Since the support of $\pi$ only contains pairs $(y,y')$ such that $d(y,y') = d(y,\mathsf{supp}(\nu'))$, we see that

$$\sum_{y,y'} \pi_{y,y'}\delta_{\mathcal{M},d(y,y')}(\varepsilon) = \sum_{y,y'} \pi_{y,y'}\delta_{\mathcal{M},d(y,\mathsf{supp}(\nu'))}(\varepsilon) = \sum_y \nu_y \delta_{\mathcal{M},d(y,\mathsf{supp}(\nu'))}(\varepsilon) \ .$$

The result follows. $\square$

## C  Proofs from Section 4

*Proof of Theorem 8.* Using the tools from Section 3, the analysis is quite straightforward. Given $x,x' \in 2^{\mathcal{U}}$ with $x \simeq_r x'$, we write $\omega = \mathcal{S}_\eta^{\mathsf{wo}}(x)$ and $\omega' = \mathcal{S}_\eta^{\mathsf{wo}}(x')$ and note that $\mathsf{TV}(\omega,\omega') = \eta$. Next we define $x_0 = x \cap x'$ and observe that either $x_0 = x$ or $x_0 = x'$ by the definition of $\simeq_r$. Let $\omega_0 = \mathcal{S}_\eta^{\mathsf{po}}(x_0)$. Then the decompositions of $\omega$ and $\omega'$ induced by their maximal coupling have either $\omega_1 = \omega_0$ when $x = x_0$ or $\omega_1' = \omega_0$ when $x' = x_0$. Noting that applying advanced joined convexity in the former case leads to an additional cancellation we see that the maximum will be attained when $x' = x_0$. In this case the distribution $\omega_1$ is given by $\omega_1(y \cup \{v\}) = \omega_0(y)$. This observation yields an obvious $d_{\simeq_r}$-compatible coupling between $\omega_1$ and $\omega_0 = \omega_1'$: first sample $y'$ from $\omega_0$ and then build $y$ by adding $v$ to $y'$. Since every pair of datasets generated by this coupling has distance one with respect to $d_{\simeq_r}$, Theorem 7 yields the bound $\delta_{\mathcal{M}'}(\varepsilon') \leq \eta\delta_{\mathcal{M}}(\varepsilon)$. $\square$

*Proof of Theorem 9.* The analysis proceeds along the lines of the previous proof. First we note that for any $x,x' \in 2^{\mathcal{U}}$ with $x \simeq_s x'$, the total variation distance between $\omega = \mathcal{S}_m^{\mathsf{wo}}(x)$ and $\omega' = \mathcal{S}_m^{\mathsf{wo}}(x')$ is given by $\eta = \mathsf{TV}(\omega,\omega') = m/n$. Applying advanced joint convexity (Theorem 2) with the decompositions $\omega = (1-\eta)\omega_0 + \eta\omega_1$ and $\omega' = (1-\eta)\omega_0 + \eta\omega_1'$ given by the maximal coupling, the analysis of $D_{e^{\varepsilon'}}(\omega M\|\omega'M)$ reduces to bounding the divergences $D_{e^\varepsilon}(\omega_1 M\|\omega_0 M)$ and $D_{e^\varepsilon}(\omega_1 M\|\omega_1'M)$. In this case both quantities can be bounded by $\delta_{\mathcal{M}}(\varepsilon)$ by constructing appropriate $d_{\simeq_s}$-compatible couplings and combining (5) with Theorem 7.

We construct the couplings as follows. Suppose $v, v' \in \mathcal{U}$ are the elements where $x$ and $x'$ differ: $x_v = x'_v + 1$ and $x'_{v'} = x_{v'} + 1$. Let $x_0 = x \cap x'$. Then we have $\omega_0 = \mathcal{S}_m^{\text{wo}}(x_0)$. Furthermore, writing $\tilde{\omega}_1 = \mathcal{S}_{m-1}^{\text{wo}}(x_0)$ we have $\omega_1(y) = \tilde{\omega}_1(y \cap x_0)$ and $\omega'_1(y) = \tilde{\omega}_1(y \cap x_0)$. Using these definitions we build a coupling $\pi_{1,1}$ between $\omega_1$ and $\omega'_1$ through the following generative process: sample $y_0$ from $\tilde{\omega}_1$ and then let $y = y_0 \cup \{v\}$ and $y' \cup \{v'\}$. Similarly, we build a coupling $\pi_{1,0}$ between $\omega_1$ and $\omega_0$ as follows: sample $y_0$ from $\tilde{\omega}_1$, sample $u$ uniformly from $x_0 \setminus y_0$, and then let $y = y_0 \cup \{v\}$ and $y' = y_0 \cup \{u\}$. It is obvious from these constructions that $\pi_{1,1}$ and $\pi_{0,1}$ are both $d_{\simeq_s}$-compatible. Plugging these observations together, we get $\delta_{\mathcal{M}'}(\varepsilon') \leq (m/n)\delta_{\mathcal{M}}(\varepsilon)$. $\qquad\square$

*Proof of Theorem 10.* To bound the privacy profile of the subsampled mechanism $\mathcal{M}^{\mathcal{S}_m^{\text{wr}}}$ on $2_n^{\mathcal{U}}$ with respect to $\simeq_s$ we start by noting that taking $x, x' \in 2_n^{\mathcal{U}}$, $x \simeq_s x'$, the total variation distance between $\omega = \mathcal{S}_m^{\text{wr}}(x)$ and $\omega' = \mathcal{S}_m^{\text{wr}}(x')$ is given by $\eta = \text{TV}(\omega, \omega') = 1 - (1 - 1/n)^m$. To define appropriate mixture components for applying the advanced joint composition property we write $v$ and $v'$ for the elements where $x$ and $x'$ differ and $x_0 = x \cap x'$ for the common part between both datasets. Then we have $\omega_0 = \mathcal{S}_m^{\text{wr}}(x_0)$. Furthermore, $\omega_1$ is the distribution obtained from sampling $\tilde{y}$ from $\tilde{\omega}_1 = \mathcal{S}_{m-1}^{\text{wr}}(x)$ and building $y$ by adding one occurrence of $v$ to $\tilde{y}$. Similarly, sampling $y'$ from $\omega'_1$ corresponds to adding $v'$ to a multiset sampled from $\mathcal{S}_{m-1}^{\text{wr}}(x')$.

Now we construct appropriate distance-compatible couplings. First we let $\pi_{1,1} \in \mathbb{P}(\mathbb{N}_m^{\mathcal{U}} \times \mathbb{N}_m^{\mathcal{U}})$ be the distribution given by sampling $y$ from $\omega_1$ as above and outputting the pair $(y, y')$ obtained by replacing each $v$ in $y$ by $v'$. It is immediate from this construction that $\pi_{1,1}$ is a $d_{\simeq_s}$-compatible coupling between $\omega_1$ and $\omega'_1$. Furthermore, using the notation from Theorem 7 and the construction of the maximal coupling, we see that for $k \geq 1$:

$$\omega_1(Y_k) = \frac{\omega(Y_k) - (1-\eta)\omega_0(Y_k)}{\eta} = \frac{\Pr_{y \sim \omega}[y_v = k]}{\eta} = \frac{1}{\eta}\binom{m}{k}\left(\frac{1}{n}\right)^k\left(1 - \frac{1}{n}\right)^{m-k},$$

where we used $\omega_0(Y_k) = 0$ since $\omega_0$ is supported on multisets that do not include $v$. Therefore, the distributions $\mu_1 = \omega_1 M$ and $\mu'_1 = \omega'_1 M$ satisfy

$$\eta D_{e^\varepsilon}(\mu_1 \| \mu'_1) \leq \sum_{k=1}^m \binom{m}{k}\left(\frac{1}{n}\right)^k\left(1 - \frac{1}{n}\right)^{m-k}\delta_{\mathcal{M},k}(\varepsilon) . \tag{7}$$

On the other hand, we can build a $d_{\simeq_s}$-compatible coupling between $\omega_1$ and $\omega_0$ by first sampling $y$ from $\omega_1$ and then replacing each occurrence of $v$ by an element picked uniformly at random from $x_0$. Again, this shows that $D_{e^\varepsilon}(\mu_1 \| \mu_0)$ is upper bounded by the right hand side of (7).

Therefore, we conclude that

$$\delta_{\mathcal{M}'}(\varepsilon') \leq \sum_{k=1}^m \binom{m}{k}\left(\frac{1}{n}\right)^k\left(1 - \frac{1}{n}\right)^{m-k}\delta_{\mathcal{M},k}(\varepsilon) .$$

$\qquad\square$

*Proof of Theorem 11.* Suppose $x \simeq_r x'$ with $|x| = n$ and $|x'| = n - 1$. This is the worst-case direction for the neighbouring relation like in the proof of Theorem 8. Let $\omega = \mathcal{S}_m^{\text{wr}}(x)$ and $\omega = \mathcal{S}_m^{\text{wr}}(x')$. We have $\eta = \text{TV}(\omega, \omega') = 1 - (1 - 1/n)^m$, and the factorization induced by the maximal coupling has $\omega_0 = \omega'_1 = \omega'$ and $\omega_1$ is given by first sampling $\tilde{y}$ from $\mathcal{S}_{m-1}^{\text{wr}}(x)$ and then producing $y$ by adding to $\tilde{y}$ a copy of the element $v$ where $x$ and $x'$ differ. This definition of $\omega_1$ suggests the following coupling between $\omega_1$ and $\omega_0$: first sample $y$ from $\omega_1$, then produce $y'$ by replacing each copy of $v$ with a element from $x'$ sampled independently and uniformly. By construction we see that this coupling is $d_{\simeq_s}$-compatible, so we can apply Theorem 7. Using the same argument as in the proof of Theorem 10 we see that $\eta\omega_1(Y_k) = \binom{m}{k}(1/n)^k(1 - 1/n)^{m-k}$. Thus, we finally get

$$D_{e^{\varepsilon'}}(\mathcal{M}^{\mathcal{S}_m^{\text{wr}}}(x) \| \mathcal{M}^{\mathcal{S}_m^{\text{wr}}}(x')) = \eta D_{e^\varepsilon}(\omega_1 M \| \omega_0 M)$$

$$\leq \eta \sum_{k=1}^m \omega_1(Y_k)\delta_{\mathcal{M},k}(\varepsilon)$$

$$= \sum_{k=1}^m \binom{m}{k}\left(\frac{1}{n}\right)^k\left(1 - \frac{1}{n}\right)^{m-k}\delta_{\mathcal{M},k}(\varepsilon) .$$

$\square$

**Theorem 14.** *Let $\mathcal{M} : 2^{\mathcal{U}} \to \mathbb{P}(Z)$ be a mechanism with privacy profile $\delta_{\mathcal{M}}$ with respect to $\simeq_s$. Then the privacy profile with respect of $\simeq_s$ of the subsampled mechanism $\mathcal{M}' = \mathcal{M}^{\mathcal{S}^{\text{po}}_\gamma} : 2^{\mathcal{U}}_n \to \mathbb{P}(Z)$ on datasets of size $n$ satisfies the following:*

$$\delta_{\mathcal{M}'}(\varepsilon') \leq \gamma\beta\delta_{\mathcal{M}}(\varepsilon) + \gamma(1-\beta)\left(\sum_{k=1}^{n-1}\tilde{\gamma}_k\delta_{\mathcal{M}}(\varepsilon_k) + \tilde{\gamma}_n\right) ,$$

*where $\varepsilon' = \log(1 + \gamma(e^\varepsilon - 1))$, $\beta = e^{\varepsilon'}/e^\varepsilon$, $\varepsilon_k = \varepsilon + \log(\frac{\gamma}{1-\gamma}(\frac{n}{k} - 1))$, and $\tilde{\gamma}_k = \binom{n-1}{k-1}\gamma^{k-1}(1-\gamma)^{n-k}$.*

*Proof of Theorem 14.* Suppose $x, x' \in 2^{\mathcal{U}}_n$ are sets of size $n$ related by the substitution relation $\simeq_s$. Let $\omega = \mathcal{S}^{\text{po}}_\eta(x)$ and $\omega' = \mathcal{S}^{\text{po}}_\eta(x')$ and note that $\text{TV}(\omega, \omega') = \eta$. Let $x_0 = x \cap x'$ and $v = x \setminus x_0$, $v' = x' \setminus x_0$. In this case the factorization induced by the maximal coupling is obtained by taking $\omega_0 = \mathcal{S}^{\text{po}}_\eta(x_0)$, $\omega_1(y \cup \{v\}) = \omega_0(y)$, and $\omega'_1(y \cup \{v'\}) = \omega_0(y)$. From this factorization we see it is easy to construct a coupling $\pi_{1,1}$ between $\omega_1$ and $\omega'_1$ that is $d_{\simeq_s}$-compatible. Therefore we have $D_{e^\varepsilon}(\omega_1 M \| \omega'_1 M) \leq \delta_{\mathcal{M}}(\varepsilon)$.

Since we have already identified that no $d_{\simeq_s}$-compatible coupling between $\omega_1$ and $\omega_0$ can exist, we shall further decompose these distributions "by hand". Let $\nu_k = \mathcal{S}^{\text{wo}}_k(x_0)$ and note that $\nu_k$ corresponds to the distribution $\omega_0$ conditioned on $|y| = k$. Similarly, we define $\tilde{\nu}_k$ as the distribution corresponding to sampling $\tilde{y}$ from $\mathcal{S}^{\text{wo}}_{k-1}(x_0)$ and outputting the set $y$ obtained by adding $v$ to $\tilde{y}$. Then $\tilde{\nu}_k$ equals the distribution of $\omega_1$ conditioned on $|y| = k$. Now we write $\gamma_k = \text{Pr}_{y\sim\omega_0}[|y| = k] = \binom{n-1}{k}\gamma^k(1-\gamma)^{n-1-k}$ and $\tilde{\gamma}_k = \text{Pr}_{y\sim\omega_1}[|y| = k] = \binom{n-1}{k-1}\gamma^{k-1}(1-\gamma)^{n-k}$. With these notations we can write the decompositions $\omega_0 = \sum_{k=0}^{n-1}\gamma_k\nu_k$ and $\omega_1 = \sum_{k=1}^{n}\tilde{\gamma}_k\tilde{\nu}_k$. Further, we observe that the construction of $\tilde{\nu}_k$ and $\nu_k$ shows there exist $d_{\simeq_s}$-compatible couplings between these pairs of distributions when $1 \leq k \leq n-1$, leading to $D_{e^\varepsilon}(\tilde{\nu}_k M \| \nu_k M) \leq \delta_{\mathcal{M}}(\varepsilon)$. To exploit this fact we first write

$$D_{e^\varepsilon}(\omega_1 M \| \omega_0 M) = D_{e^\varepsilon}\left(\sum_{k=1}^{n-1}\tilde{\gamma}_k\tilde{\nu}_k M + \tilde{\gamma}_n\tilde{\nu}_n M \,\middle\|\, \gamma_0\nu_0 M + \sum_{k=1}^{n-1}\gamma_k\nu_k M\right) .$$

Now we use that $\alpha$-divergences can be applied to arbitrary non-negative measures, which are not necessarily probability measures, using the same definition we have used so far. Under this relaxation, given non-negative measures $\nu_i, \nu'_i$, $i = 1, 2$, on a measure space $Z$ we have $D_\alpha(\nu_1 + \nu_2 \| \nu'_1 + \nu'_2) \leq D_\alpha(\nu_1 \| \nu'_1) + D_\alpha(\nu_2 \| \nu'_2)$, $D_\alpha(a\nu_1 \| b\nu_2) = aD_{\alpha b/a}(\nu_1 \| \nu_2)$ for $a \geq 0$ and $b > 0$, and $D_\alpha(\nu_1 \| 0) = \nu_1(Z)$. Using these properties on the decomposition above we see that

$$D_{e^\varepsilon}(\omega_1 M \| \omega_0 M) \leq \sum_{k=1}^{n-1}\tilde{\gamma}_k D_{e^{\varepsilon_k}}(\tilde{\nu}_k M \| \nu_k M) + \tilde{\gamma}_n$$

$$\leq \sum_{k=1}^{n-1}\tilde{\gamma}_k\delta_{\mathcal{M}}(\varepsilon_k) + \tilde{\gamma}_n ,$$

where $e^{\varepsilon_k} = (\gamma_k/\tilde{\gamma}_k)e^\varepsilon = (\gamma/(1-\gamma))(n/k - 1)e^\varepsilon$. $\square$

# D Proofs from Section 5

*Proof of Lemma 12.* We start by observing that for any $x \in X$ the distribution $\mu = \mathcal{M}^{\mathcal{S}}_{v,p}(x)$ must be a mixture $\mu = (1-\theta)\nu_0 + \theta\nu_1$ for some $\theta \in [0, 1]$. This follows from the fact that there are only two possibilities $\nu_0$ and $\nu_1$ for $\mathcal{M}_{v,p}(y)$ depending on whether $v \notin y$ or $v \in y$. Similarly, taking $x \simeq_X x'$ we get $\mu' = \mathcal{M}^{\mathcal{S}}_{v,p}(x')$ with $\mu' = (1-\theta')\nu_0 + \theta'\nu_1$ for some $\theta' \in [0, 1]$. Assuming (without loss of generality) $\theta \geq \theta'$, we use the advanced joint convexity property of $D_\alpha$ to get

$$D_{e^{\varepsilon'}}(\mu \| \mu') = \theta D_{e^\varepsilon}(\nu_1 \| (1 - \theta'/\theta)\nu_0 + (\theta'/\theta)\nu_1)$$
$$\leq \theta(1 - \theta'/\theta)D_{e^\varepsilon}(\nu_1 \| \nu_0) = (\theta - \theta')\psi_p(\varepsilon) \leq \theta\psi_p(\varepsilon) ,$$

where $\varepsilon' = \log(1 + \theta(e^\varepsilon - 1))$ and $\beta = e^{\varepsilon'}/e^\varepsilon$, and the inequality follows from joint convexity. Now note the inequalities above are in fact equalities when $\theta' = 0$, which is equivalent to the fact $v \notin x'$ because $\mathcal{S}$ is a natural subsampling mechanism. Thus, observing that the function $\theta \mapsto \theta\psi_p(\log(1 + (e^{\varepsilon'} - 1)/\theta))$ is monotonically increasing, we get

$$\sup_{x \simeq_X x'} D_{e^{\varepsilon'}}(\mathcal{M}_{v,p}^{\mathcal{S}}(x)\|\mathcal{M}_{v,p}^{\mathcal{S}}(x')) = \sup_{x \simeq_X x', v \notin x'} \theta\psi_p(\log(1 + (e^{\varepsilon'} - 1)/\theta))$$

$$= \eta\psi_p(\log(1 + (e^{\varepsilon'} - 1)/\eta)) = \eta\psi_p(\varepsilon) \ .$$

$\square$