[Reviews · NeurIPS 2018]

Reviewer 1



This paper carefully studies the privacy amplification effect of subsampling in differential privacy. That is, when a differentially private algorithm is run on a random subset of the whole dataset, the privacy guarantee for the whole dataset is amplified relative to that of the subset. This is particularly important for analyzing private versions of algorithms like stochastic gradient descent. This is a well-known and important property, but this paper gives a more thorough analysis than has previously been presented. Previously this property has only been analyzed "as needed" and, to my knowledge, no extensive study is available. I think the tight bounds in this paper will be a useful reference and the techniques may find other applications. My only complaint about the paper is that it is very technical and hard for a non-expert to digest. I suggest including easily-interpretable self-contained statements of the main results that do not rely on all the notation developed. I.e., make it as easy as possible to cite a result from this paper using standard notation.

Reviewer 2



This paper derives differential privacy guarantees for algorithms that combines subsampling and differentially private mechanisms. It first presents some general tools, and then applies it to different subsampling methods. Analyzing the privacy amplification of subsampling is a very important and interesting problem in differentially private algorithm design. And this paper provides results on different sampling methods and algorithms with different privacy guarantees. More comparison with previous work would make the result more solid, and the presentation might be improved a bit for better readability. More detailed comments: - It is important to mention related work which proves privacy bound of privacy amplification with sampling in more detail. For example, [1] provides a bound for Gaussian mechanism with sampling. How does the result compare? It might be good to illustrate the comparison with figures and values in some real scenarios. - The presentation in Section 3 might be more clear with subsections for divergences and privacy profiles. - I think Section 4 is an important (if not the most important) section of the paper. So giving it a more informative name may help emphasize its importance and the significance of the paper. Small typo: - Line 11: there are two "introduces" - Line 114: is the "increasing" supposed to be "decreasing"? Otherwise this couldn't be compatible with limit equal to 0.

Reviewer 3



The manuscript is a contribution to the analysis of one of most popular tools for differential privacy, namely subsampling. Although many of the results for upper bounds of subsampling schemes already exist in the literature, a general framework that not only recovers them but also can asses their tightness is still valuable. -Page 7, line 298: "Note that this setting cannot be implemented using sampling without replacement since under the remove/add relation we cannot a priori guarantee that the input dataset will have at least size m because the size of the dataset must be kept private." This sentence should be rephrased and the discussion should be elaborated. Also, it is not clear why the size of the dataset must be kept private and why this implies that we cannot a priori guarantee that the input dataset will have at least size m. - Page 3, line 98: There may be a notational issue in the build-up of M^S: In M(S(x)), S(x) is viewed as a subset although S is defined a function that maps a database to a probability distribution. - Below Thm1, line 144 , propoerty (ii): Should it be “monotonically decreasing”? - Page 4, line 156: typo: change v_1 to \pi_1. - Page 7, line 257: change "results" to "result".